# Characterization of the Regulatory Network under Waterlogging Stress in Soybean Roots via Transcriptome Analysis

**DOI:** 10.3390/plants13182538

**Published:** 2024-09-10

**Authors:** Yo-Han Yoo, Seung-Yeon Cho, Inhye Lee, Namgeol Kim, Seuk-Ki Lee, Kwang-Soo Cho, Eun Young Kim, Ki-Hong Jung, Woo-Jong Hong

**Affiliations:** 1Central Area Crop Breeding Division, Department of Central Area Crop Science, National Institute of Crop Science, Rural Development Administration, Suwon 16429, Republic of Korea; yohan04@korea.kr (Y.-H.Y.); ih22@korea.kr (I.L.); qoo2010@korea.kr (N.K.); sklee77@korea.kr (S.-K.L.); kscholove@korea.kr (K.-S.C.); 2Department of Smart Farm Science, Kyung Hee University, Yongin 17104, Republic of Korea; jenny6546@khu.ac.kr (S.-Y.C.); key0620@khu.ac.kr (E.Y.K.); 3Graduate School of Green Bio-Science & Crop Biotech Institute, Kyung Hee University, Yongin 17104, Republic of Korea; khjung2010@khu.ac.kr

**Keywords:** RNA-seq, soybean, transcriptome analysis, transcription factor, waterlogging stress

## Abstract

Flooding stress caused by climate change is a serious threat to crop productivity. To enhance our understanding of flooding stress in soybean, we analyzed the transcriptome of the roots of soybean plants after waterlogging treatment for 10 days at the V2 growth stage. Through RNA sequencing analysis, 870 upregulated and 1129 downregulated differentially expressed genes (DEGs) were identified and characterized using Gene Ontology (GO) and MapMan software (version 3.6.0RC1). In the functional classification analysis, “alcohol biosynthetic process” was the most significantly enriched GO term in downregulated DEGs, and phytohormone-related genes such as ABA, cytokinin, and gibberellin were upregulated. Among the transcription factors (TFs) in DEGs, AP2/ERFs were the most abundant. Furthermore, our DEGs encompassed eight soybean orthologs from Arabidopsis and rice, such as 1-aminocyclopropane-1-carboxylate oxidase. Along with a co-functional network consisting of the TF and orthologs, the expression changes of those genes were tested in a waterlogging-resistant cultivar, PI567343. These findings contribute to the identification of candidate genes for waterlogging tolerance in soybean, which can enhance our understanding of waterlogging tolerance.

## 1. Introduction

Soybean (*Glycine max* L. [Merril]) is valued as one of the most economically significant oilseed crops as it contains an abundance of edible protein and oil [1]. Soybean plays a pivotal role in various products and services, including human food, animal feed, biofuel production, and many others, as an essential legume crop. Global soybean production has increased over time, reaching 320.2 million metric tons in 2015, from 17 million metric tons in 1960. According to the United States Department of Agriculture’s agricultural outlook for 2025, the global soybean trade volume is expected to increase by 22% and the soybean oil trade volume is expected to increase by 30% [2]. However, recent changes in precipitation due to global warming have had a negative effect on the yield of soybean, which is highly sensitive to waterlogging stress [3].

Plants exhibit a variety of physiological effects under water stress. First, roots that are not sufficiently supplied with oxygen inhibit growth and promote the formation of aerenchyma and adventitious roots [4]. Additionally, ethylene hormone accumulates inside the plant, inducing stress responses in the leaves and roots, and the chlorosis phenomenon is observed in the leaves [5,6]. Finally, the metabolic pathway shifts from respiration to fermentation, resulting in lower energy productivity. Consequently, this leads to inhibited plant growth, reduced growth rates of stems and leaves, delayed flowering, and decreased yield [7]. Indeed, temporary waterlogging has detrimental effects on soybean, leading to stunted plant growth, impaired development, and decreased seed production at critical stages of the life cycle, including germination, vegetative growth, and reproductive phases [8,9].

Rapid changes in climate conditions are increasing the frequency of repeated flooding in agricultural areas around the world [10]. These changes can cause substantial harm to crop yields and impact global food security. For example, only two days of flooding during the late vegetative stage can result in an 18% loss in yield, with a potential increase to 26% if flooding occurs during the early reproductive stage of soybean [11]. In addition, flood-related damage and control costs in the United States in 2016 totaled USD 217 million, 3.4 times more than for drought (https://legacy.rma.usda.gov/data/cause.html; accessed on 11 March 2024). In some developing countries, inadequate draining systems may lead to frequent flooding, resulting in poverty and food insecurity [12].

The QTL analysis of soybean populations has revealed the genomic regions associated with flooding tolerance [13,14]. Briefly, a single QTL associated with the simple sequence repeat marker Sat_064 was identified from two recombinant inbred populations, namely “Archer” × “Minsoy” and “Archer” × “Noir I”. Under waterlogging stress, the allele from Archer enhanced plant growth by 11–18% and yield by 47–180%. Furthermore, two markers were found to be related to flooding tolerance, Satt385 on chromosome 5 and Satt269 on chromosome 13, and four markers, Satt59, Satt160, Satt252, and Satt485, were identified as QTLs related to flooding tolerance [12]. Unfortunately, the understanding of gene function related to waterlogging stress in soybean is currently limited; only a few quantitative trait loci (QTLs) associated with waterlogging tolerance have been identified [15,16].

The development of next-generation sequencing technology has facilitated the analysis of interactions between plants and abiotic stress at the transcriptome level [17,18]. Specifically, RNA sequencing (RNA-seq) was conducted in various crops, such as Arabidopsis, rice, maize, and pea, with the aim of elucidating the molecular mechanisms and signaling responses implicated in flooding stress [19,20,21]. In general, two types of flooding are encountered in the field: (1) waterlogging, in which the roots and some portions of the shoot are submerged, and (2) complete submergence, in which the entire plant is submerged underwater. To date, most of transcriptome analysis has focused on complete submergence treatment even though waterlogging is more frequently seen in the field [22,23].

Here, we subjected Daewonkong, a cultivar of soybean (*Glycine max* L. [Merrill]), to waterlogging stress to investigate transcriptome variations and elucidate genetic resources for waterlogging stress-resistant soybean. The result of our transcriptome analysis suggests 1999 differentially expressed genes that could be a foundation for soybean breeding for a waterlogging-resistant variety combined with QTL analysis.

## 2. Materials and Methods

### 2.1. Experimental Design

We conducted waterlogging stress treatment on Daewonkong, a cultivar of soybean (*Glycine max* L. [Merrill]). We aimed to investigate the transcriptome changes following waterlogging by comparing the roots of control and stress-treated plants. Gene Ontology (GO) enrichment and MapMan analysis were performed on selected differentially expressed genes (DEGs) using gene set annotations. Then, we searched for soybean orthologs of genes associated with waterlogging tolerance in other crops, such as Arabidopsis and rice, and compared the identified genes with the DEGs. In addition, co-functional network analysis was performed using major transcription factors and orthologs. Lastly, differences in the expression of key candidate genes predicted in the network were confirmed using a waterlogging-tolerant cultivar. The overall study workflow is summarized in Appendix A.

### 2.2. Plant Materials and Waterlogging Stress Treatments

Plants of the soybean (*Glycine max* L. [Merrill]) cv. Daewonkong were grown in plastic pots in an incubator until they reached the V2 stage (14 h light/10 h dark, 28 °C (day)/24 °C (night), humidity 60%). Based on the previous reports [24,25] and our morphological observations (Figure 1A–C), an appropriate water level to submerge the soil was maintained for 10 days to apply waterlogging stress, followed by a recovery period of 15 days. To determine the physiological features affected by waterlogging stress treatment, the roots were analyzed at three time points: before stress treatment, after treatment, and after recovery.

### 2.3. RNA Sequencing (RNA-Seq) Analysis

Roots were collected from control (control) and 10 days of waterlogging stress-treated plants (WS) used in three replicated experiments and immediately frozen in liquid nitrogen. Total RNA was extracted using the RNeasy Plant Mini Kit (Qiagen, Hilden, Germany), and the transcripts for library construction were selected based on the RIN value (RIN > 7). The RNA-Seq library was constructed using the TruSeq Stranded mRNA Library Prep kit in accordance with the manufacturer’s instructions (TruSeq Stranded mRNA Reference Guide #1000000040498v00 library) and sequenced using the Illumina NovaSeq 6000 platform by Macrogen Inc.

The raw data were trimmed using Trimmomatic [26] with the following parameters: window size = 4, mean quality = 15, and min length = 36. The cleaned reads were aligned to a soybean reference genome (Glycine_max_v2.1) using HISAT2 software (version 2.1.0) [27]. Read counts were calculated using StringTie software [28] with the option of fragments per kilobase of transcript per million mapped reads (FPKM). Lastly, differentially expressed genes (DEGs) were statistically elucidated after quantile normalization and an independent *t*-test using the following criteria: average FPKM of each sample group ≥ 4 to filter expressed genes, |fold change (WS/Control)| > 2, and *p*-value < 0.05.

### 2.4. Functional Analysis of DEGs Using Gene Ontology (GO) and MapMan Software

The GO terms of the DEGs were assigned via the AgriGOv2 toolkit [29] (http://systemsbiology.cau.edu.cn/agriGOv2/classification_analysis.php?category=Plant&&family=Fabaceae). The GO terms with a fold enrichment value of >2 and FDR < 0.05 were considered as enriched terms. The results were illustrated using the R package ggplot2 [30]. Functional classification of the DEGs was performed using MapMan software with an X4.4 Glycine max mapping file [31].

### 2.5. Ortholog Analysis

Previously reported waterlogging-related genes were retrieved from the literature, including those in rice and Arabidopsis [32,33]. To match locus information for the genes, the funRiceGenes database (https://funricegenes.github.io/) and the NCBI database (https://www.ncbi.nlm.nih.gov/) were utilized. Lastly, the locus information of rice and Arabidopsis was mapped to the soybean locus using the g:Orth toolkit [34].

### 2.6. Co-Functional Networks Analysis

Using the SoyNet tool (http://www.inetbio.org/soynet) [35], we generated a functional gene network involving various transcription factors (TFs) and soybean orthologs of flooding stress-related genes identified in other crops. The network was edited using the Cytoscape tool (version 3.9.1) [36].

### 2.7. RNA Extraction and Quantitative Real-Time PCR (qRT-PCR) Analysis

Samples from the roots of Daewonkong (control) and PI 567343 (waterlogging-resistant variety) treated with waterlogging stress for 10 days from 2-week-old soy seedlings (V2 stage) were collected and immediately frozen in liquid nitrogen. Total RNA was extracted using the RNeasy Plant Mini Kit in accordance with the manufacturer’s protocol (Qiagen, Hilden, Germany). First-strand cDNA was synthesized using M-MuLV Reverse Transcriptase (Thermo Fisher Scientific, Waltham, MA, USA) and the oligo(dT)-18 primer. Then, qPCR was performed using a QuantStudio 5 Real-Time PCR instrument (Thermo Fisher Scientific, Waltham, MA, USA). To normalize the amplified transcripts, we used a primer pair for soybean Actin11 gene (*Act11/Glyma18g52780.1*) [37]. Student’s *t*-test was used for the statistical analysis of the results. All primers for the genes used in these analyses are presented in Appendix A.

## 3. Results and Discussion

### 3.1. Physiological Responses of Soybean Roots Exposed to a Waterlogging Stress

Two-week-old soybean seedlings (V2 stage) grown in an incubator were exposed to waterlogging stress for 10 days (Figure 1A,B) and then grown for a further 15 days under normal conditions as a recovery treatment (Figure 1C). Observing the roots at each stage revealed that root development was inhibited in the waterlogging-treated plants and adventitious roots (ARs) were increased (Figure 1B). The inhibition of root development caused by waterlogging stress was not resolved, even after the 15-day recovery period (Figure 1C).

Plants promote the development of ARs to smoothly facilitate the uptake of nutrients and the transport of gas under waterlogging stress [38]. ARs can be categorized into various types, including hypocotyl roots, crown roots, brace roots, nodal roots, stem roots, junction roots, and prop roots, depending on physical characteristics and induction conditions [39]. In soybean, waterlogging stress is predicted to induce hypocotyl roots (Figure 1B).

The development of ARs during waterlogging stress has been studied in various crops. For example, ARs gradually formed at the hypocotyl after three days of partial submergence owing to the action of ethylene and auxin in tomato [40]. In maize, waterlogging stress increases the development of crown roots, with more crown roots observed in waterlogging-tolerant lines than in waterlogging-sensitive lines [41]. Cucumber also promotes the development of ARs under waterlogging stress by ethylene and auxin [42]. These findings suggest that the development of plant ARs is primarily governed by the interplay of ethylene and auxin during waterlogging stress, although their specific roles can differ across plant species.

### 3.2. Identification of Differentially Expressed Genes (DEGs) under Waterlogging Stress by RNA-Seq

To identify the DEGs in response to waterlogging treatment, we performed transcriptome analysis of the roots of control and waterlogging-treated plants (10 days of waterlogging stress, Figure 1B) and identified 870 upregulated and 1129 downregulated genes in the waterlogging-treated plants relative to the control (*p*-value < 0.05 and |log2 fold-change| > 2; Figure 1D, Appendix A).

### 3.3. Gene Ontology (GO) Enrichment Reveals Biological Processes Associated with Waterlogging Stress in Soybean Roots

To explore the biological functions of the 870 upregulated and 1129 downregulated DEGs resulting from waterlogging stress in soybean roots, we investigated the GO terms of those genes within the “biological process” category. Twenty-one GO terms were highly over-represented in the group of upregulated genes and fifteen GO terms were significantly enriched in the group of downregulated genes (Figure 2, Appendix A). Lastly, eight terms were highly over-represented in both the upregulated and downregulated groups of genes.

Of the identified terms, “alcohol biosynthetic process” (20.6-fold enrichment) was the most significantly enriched term in the downregulated gene group. Alcoholic fermentation, also known as ethanol fermentation, is a biological process that converts sugars into cellular energy while generating the by-products of ethanol and carbon dioxide [43].

In plants, respiration using oxygen plays a major role under normal oxygen conditions (aerobic conditions), while fermentation is used to obtain energy under anaerobic conditions [44,45]. That is, plants produce ATP through fermentation at low efficiency during waterlogging stress, with ethanol being the main product in the process.

Interestingly, GO enrichment analysis showed that genes associated with alcohol biosynthesis were downregulated. These results might be due to the plant material used in this study, V2 stage soybean plants, unlike previous reports that usually used germinating seeds or young seedlings [25,46,47]. As shown in Figure 1B, we conducted a 10-day flooding treatment during the V2 stage and compared the results to the control group, and a significant reduction in root growth was observed. Therefore, it is predicted that prolonged flooding caused cellular damage, leading to a decrease in overall metabolic activity and a subsequent reduction in ethanol biosynthesis. The fact that the second most downregulated gene group in the GO enrichment analysis is associated with the “secondary metabolic process” further supports this prediction. Since the gene expression of waterlogging-responsive genes differs in tissues and crop species [46], our transcriptome results suggest the necessity of further research on waterlogging stress in soybean in the context of developmental stages and tissue types.

### 3.4. MapMan Analysis of Waterlogging Stress-Related Genes in Soybean Roots

The MapMan program is a powerful tool that can effectively visualize transcriptomic data and provide meaningful insights from different perspectives [48]. Therefore, we uploaded fold-change data and locus IDs for the 870 upregulated and 1129 downregulated genes for analysis by various overviews within the MapMan program (Figure 3; Appendix A). In the metabolism overview, we identified 41 elements downregulated in “fermentation,” which was consistent with the identification of the “alcohol biosynthetic process” as the most enriched biological process in the GO enrichment analysis (green box in Figure 3A).

In the cellular response overview, it was shown that the responses of thioredoxin (18 elements), ascorbate peroxidase (APX)/glutathione (16), and glutaredoxin (22) were clearly related to waterlogging stress (green box in Figure 3B, Appendix A). Plants accumulate harmful substances, such as reactive oxygen species (ROS) and malondialdehyde (MDA), internally during flooding stress [49,50]. To overcome this situation, plants enhance the activation of antioxidant enzymes and generate various compounds that can actively participate in redox reactions. For example, the overexpression of APX cDNA from eggplant in rice resulted in a ninefold increase in APX activity under flooding compared with that in non-transgenic plants [51]. The exogenous application of GSH (reduced glutathione) to sesame plants under waterlogging stress reduced the accumulation of ROS and diminished membrane damage, resulting in improved photosynthetic efficiency and biomass [52]. Our results are consistent with previous reports that redox-related genes are upregulated during waterlogging stress.

### 3.5. Expression Analysis of TFs in DEGs

Among the 1999 DEGs, we found 33 APETALA2/ethylene-responsive (AP2/ERF, upregulated: 11, downregulated: 22), 13 basic helix-loop-helix (bHLH, up: 6, down: 7), 5 basic leucine zipper (bZIP, up: 4, down: 1), 6 Cys2His2 (C2H2, up: 1, down: 5) zinc finger, 7 GRAS (up: 1, down: 6), 6 homeodomain-leucine zipper (HD-ZIP, up: 3, down: 3), 18 NAM, ATAF1/2, and CUC2 (NAC, up: 1, down: 17), 14 WRKY (up: 3, down: 11), 3 auxin response factor (ARF, down: 3), 5 heat shock factor (down: 5), and 17 myeloblastosis (MYB, down 17) transcription factor (TF) DEGs for waterlogging stress (Figure 4A, Appendix A). Interestingly, there were more than twice as many TFs in the downregulated gene group than in the upregulated gene group.

To validate our transcriptome analysis, we selected 10 TF genes (4 highly changed and 6 randomly chosen TFs) from each of the two DEGs and analyzed their expression patterns. As anticipated, the expression of the chosen TF genes was consistent with their expression pattern in the RNA-seq analysis (Figure 4B and Appendix A). We confirmed that AP2/ERF genes were the most common TFs. In particular, Group VII ethylene response factors (ERF-VIIs) are known to be involved in plant tolerance to waterlogging stress [53]. For example, in Arabidopsis, the ERF-VII genes *HRE1*, *HRE2*, *RAP2.2*, *RAP2.3*, and *RAP2.12* have been identified as pivotal regulators of flooding and low oxygen tolerance [54,55]. Two ERF-VII genes, *SNORKEL1* and *SNORKEL2*, found in deep-water rice, contributed to the elongation of the internodes to prevent the top leaves from submergence [56]. In maize, *ZmEREB180*, an ERF-VII member, has been reported to increase the formation of ARs and promote antioxidant ability during waterlogging stress [57]. The role of ERF-VII has also been studied in waterlogging-tolerant wheat (*Triticum aestivum* L.) and the dicot species Rumex and Rorippa [58,59].

The most significantly up- or downregulated TFs were a bHLH TF, *Glyma.01G129700,* and an NAC TF, *Glyma.12G221500,* respectively. Although these genes have not been characterized in soybean, some reports in pepper and Arabidopsis indicate that bHLH and NAC TFs have specific roles under waterlogging stress [60,61]. The overexpression of *CabHLH18* in pepper showed enhanced waterlogging tolerance and the Arabidopsis NAC TF, SPEEDY HYPONASTIC GROWTH, governs waterlogging-induced leaf movement by interacting with 1-aminocyclopropane-1-carboxylate (ACC) oxidase 5. Furthermore, WRKY [62], ARF [63], MYB [64], and bZIP [65] TFs have been found to be associated with waterlogging stress. Collectively, these findings suggest a strong correlation between waterlogging and various TFs in soybean.

### 3.6. Ortholog Analysis to Elucidate Flooding Stress-Related Genes in Soybean

We performed a literature search to determine whether the DEGs identified in our RNA-seq analysis encompassed previously characterized genes related to waterlogging stress. To date, although no genes have been reported in soybean, the functions of genes involved in flooding stress have been revealed in various crops, including Arabidopsis, rice, maize, and barley [32,33]. In the case of rice, we were able to find 32 submergence-related genes reported to date through the funRiceGenes database (https://funricegenes.github.io/tags/) [66].

Consequently, we identified soybean orthologs by mapping genes studied in various crops, resulting in the discovery of eight soybean genes through comparison with the DEGs list (Table 1). In brief, there were two downregulated orthologs in soybean (*Glyma.15G112700* and *Glyma.09G008400*) of an important factor in ethylene biosynthesis in Arabidopsis, *ACO* (*AT1G12010*) [67]. Furthermore, there was one downregulated ortholog (*Glyma.02G034000*) of rice mitochondrial aldehyde dehydrogenase that metabolizes acetylaldehyde after submergence [68]. These results provide insights into the molecular mechanism underpinning waterlogging stress in soybean.

### 3.7. Analysis of Co-Functional Networks of TFs and Orthologs

Transcription factors (TFs) play a crucial role in regulating the response and resistance of plants to flooding stress. Orthologs of genes in other crops reported to function in relation to flooding stress are promising candidate genes for research on waterlogging tolerance in soybeans. Among the 1999 genes with expression changed by 10 days of waterlogging stress treatment, we identified 127 TFs and 8 ortholog genes (Figure 4, Table 1). By gaining an understanding of the co-expression relationship between them, novel strategies can be explored to enhance tolerance to waterlogging stress in soybean. Consequently, we generated a co-functional gene network for the 127 TFs and 8 orthologs based on the SoyNet webtool [35]. The network was refined using the Cytoscape tool, resulting in eight TF categories being utilized as queries: 19 AP2/ERF (white nodes), 14 NAC (green nodes), 9 WRKY (light blue nodes), 9 MYB (dark blue nodes), 4 C2H2 (ZF, yellow nodes), 1 bHLH (pink node), 1 bZIP (orange node), and 2 orthologs (gray nodes). Finally, we gathered elements with more than five lines into one node and grouped them with a red dotted line (Figure 5A).

### 3.8. Stress Treatment of Waterlogging-Resistant Varieties and Quantitative Real-Time PCR (qRT-PCR) Analysis

We predicted that the 26 genes included in the red dotted line in the co-functional networks would be candidate genes that may be useful in waterlogging tolerance research (Figure 5A). To evaluate the network, we investigated the expression of nine waterlogging-responsive genes in waterlogging-resistant varieties compared with Daewonkong. The waterlogging-resistant cultivar utilized, PI 567343, is one of the germplasm lines (*G. max*) identified as a potential donor for breeding aimed at enhancing flooding tolerance (purchased from National Agrobiodiversity Center of Rural Development Administration, Republic of Korea) [73].

As for the RNA-seq sample preparation, qRT-PCR analyses were performed using the roots of the two cultivars Daewonkong and PI 567343 grown to the V2 stage and treated with waterlogging stress for 10 days. Nine genes were found to be expressed more than twice as strongly in PI 567343 than in Daewonkong. Interestingly, eight out of the nine genes were AP2/ERF TFs: the expression of *Glyma.19G213100* was confirmed to be 8-fold higher and the expression of *Glyma.17G131800* and *Glyma.10G239300* was 4-fold higher (Figure 5B).

Three AP2/ERFs in the network (*Glyma.04G041200*, *Glyma.15G180000*, and *Glyma.17G131900*) showed different expression patterns in two cultivars under waterlogging stress (Figure 4B, Figure 5B and Appendix A). Although these AP2/ERFs appear to have a negative role under waterlogging stress due to their decreased expression, the actual mode of action needs to be characterized through further analysis. In our previous study in rice, the knockout of *OsPhyB*, which was upregulated under drought stress, showed a tolerant phenotype [17]. Also, it suggests that genes in the co-functional network under abiotic stress could be co-regulated, as previously reported [74].

As mentioned in Section 3.5, these results suggest that among various TFs, AP2/ERF might be an important component in studying soybean waterlogging resistance.

## 4. Conclusions

Recent extreme climate changes have increased the severity of flooding stress, which threatens the stability of crop production. In particular, soybean, which is an economically important oilseed crop, but very sensitive to waterlogging stress urgently needs research about improving stress resistance. In this study, we investigated transcriptome variation in soybean root to investigate the molecular mechanism associated with waterlogging stress and suggested 1999 DEGs and a co-functional network.

Recently, the focus of flooding stress research has gradually shifted from the screening of major genetic loci through genome-wide association study or map-based cloning techniques to functional studies using gene editing tools such as the CRISPR/Cas9 system. In line with this trend, the candidate genes from our research could be a valuable resource to generate a waterlogging resistance variety.

## Figures and Tables

**Figure 1 plants-13-02538-f001:**
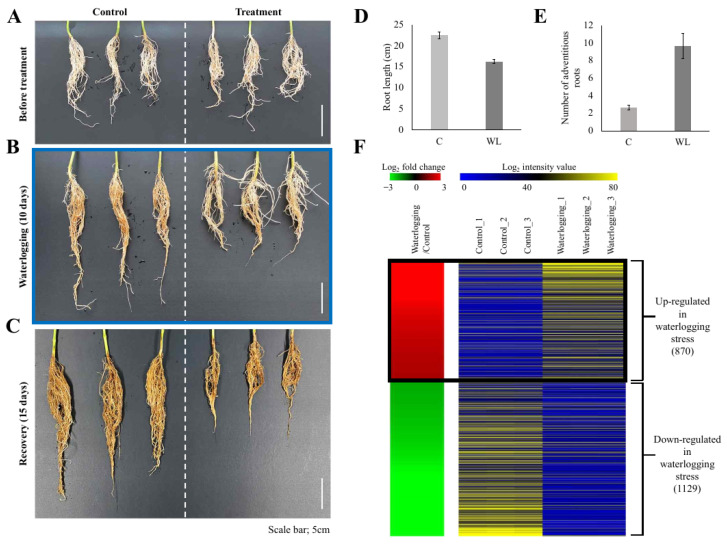
Physiological responses of soybean roots to waterlogging stress and heat-map of differentially expressed genes. Plants at the V2 stage were exposed to waterlogging stress for 10 days and allowed to recover for 15 days (**A**–**C**). Each of the images indicates before waterlogging (**A**), after 10 days of waterlogging (**B**), and 15 days of recovery (**C**). Scale bar = 5 cm. N = 3 for (**A**–**C**). Quantitative data of the root length (**D**) and the number of adventitious roots (**E**) are illustrated. Genes differentially expressed in roots during waterlogging stress were identified (**F**). Processing RNA-seq data under the criteria of FPKM > 4, *p*-value < 0.05, and |log2(fold change)| over 2 for roots exposed to waterlogging vs. mock-treated soybean roots (control) identified 1999 differentially expressed genes (DEGs). In the left panel, red indicates upregulation in the waterlogging/control comparison and green indicates downregulation in the waterlogging/control comparison. The right panel shows the average normalized FPKM values from RNA-seq experiments; blue indicates the lowest expression level and yellow indicates the highest level. Detailed data for the RNA-seq analysis are presented in Appendix A.

**Figure 2 plants-13-02538-f002:**
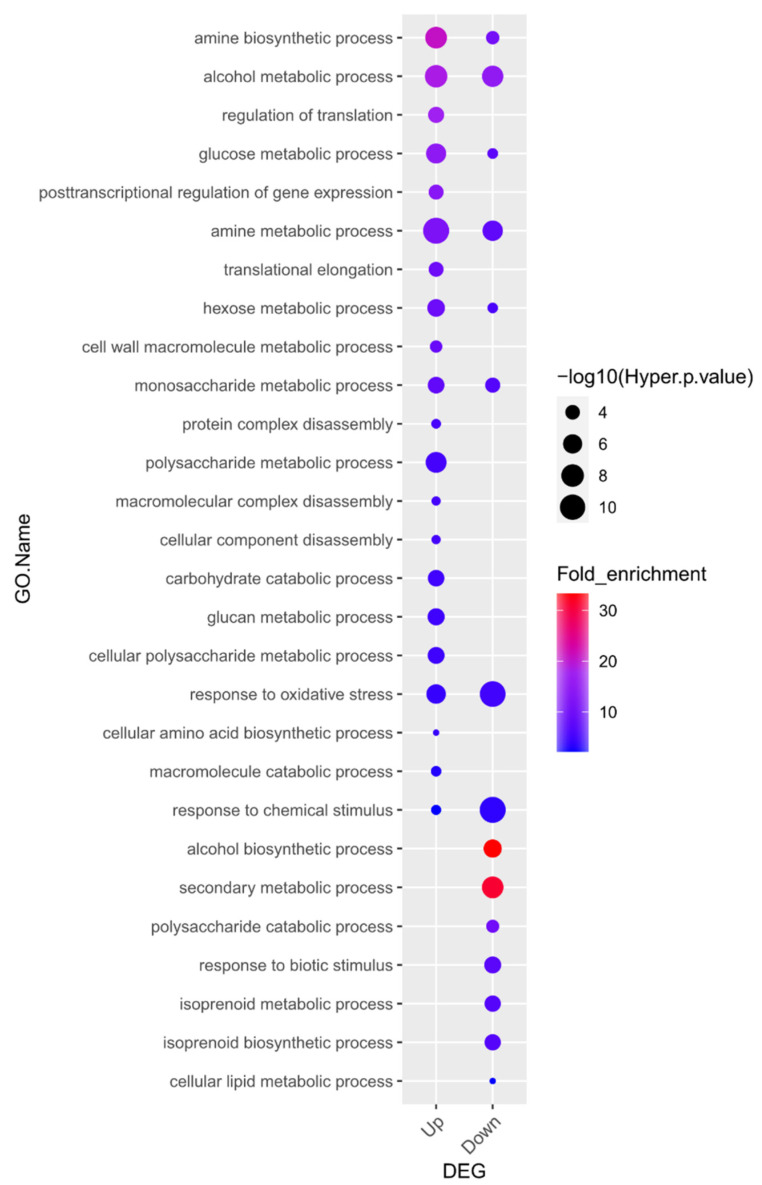
Gene Ontology (GO) enrichment analysis in the “biological process” category for genes up- and downregulated in response to waterlogging. Overall, 21 GO terms were highly over-represented, and in the downregulated gene group, 15 GO terms were significantly enriched (*p* < 0.05 and fold-enrichment values of >2 log2-fold). Details of the GO assignments are presented in Appendix A.

**Figure 3 plants-13-02538-f003:**
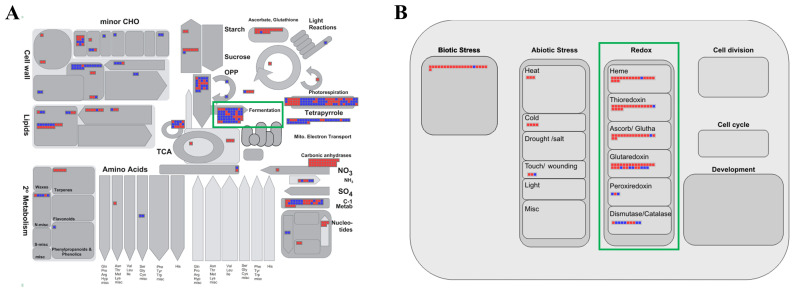
MapMan analysis of genes associated with the response to waterlogging. Overviews: (**A**) metabolism; (**B**) cellular response. Red and blue boxes indicate up- and downregulated genes, respectively; green boxes highlight the pathways related to waterlogging stress response. Detailed information is presented in Appendix A.

**Figure 4 plants-13-02538-f004:**
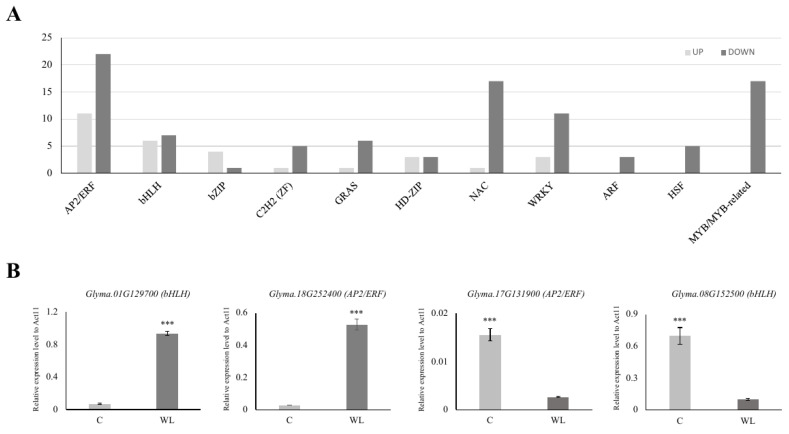
Expression analysis of major transcription factors in DEGs. Among the transcription factors, the number of upregulated genes is presented in light gray, and the number of downregulated genes is presented in dark gray (**A**). The expression of four randomly selected genes from the up- or downregulated genes was compared (**B**). C: control (light gray), WL: waterlogging (dark gray). *** *p* < 0.001.

**Figure 5 plants-13-02538-f005:**
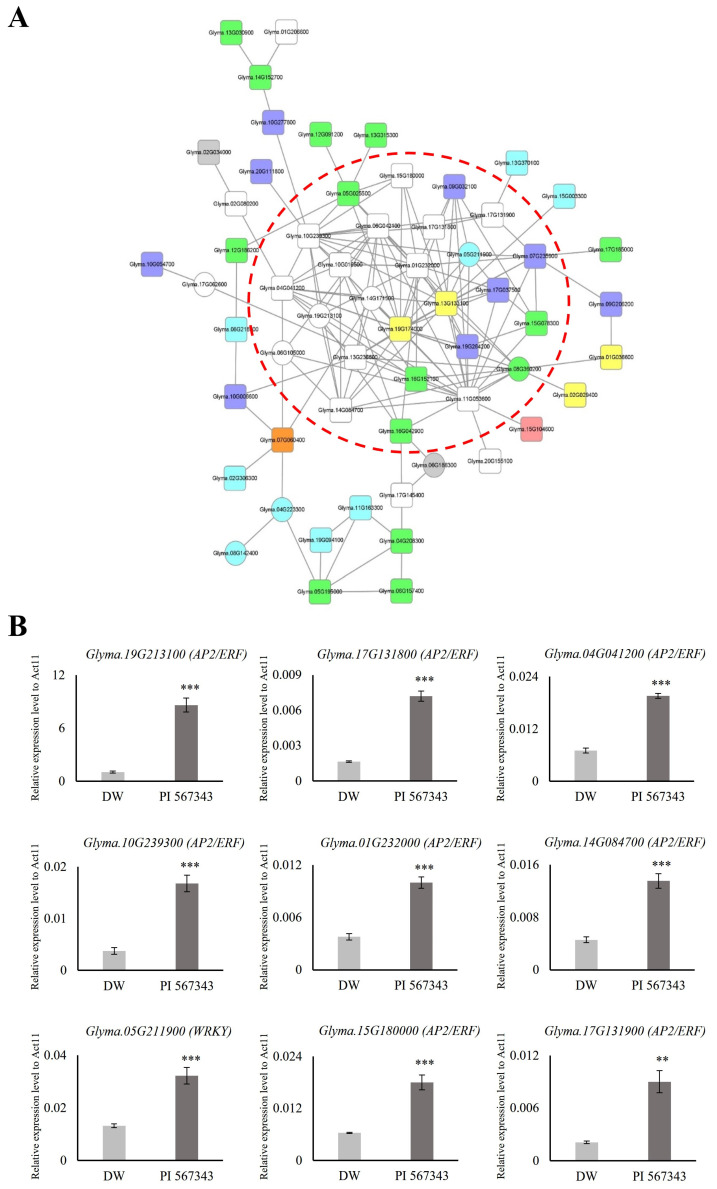
Construction and expression validation of co-functional networks associated with transcription factors regulated under waterlogging stress. (**A**) Co-functional network consists of various transcription factors retrieved from SoyNet. There are 19 AP2/ERF (white nodes), 1 bHLH (pink node), 1 bZIP (orange node), 4 C2H2 (ZF, yellow nodes), 14 NAC (green nodes), 9 WRKY (light blue nodes), 9 MYB (dark blue nodes), and 2 ortholog (gray nodes) in the network. Circular nodes represent upregulated genes; square nodes represent downregulated genes. The raw networks initially created in SoyNet are shown in Appendix A. (**B**) The expression of nine TF genes was significantly upregulated in a resistant variety, PI 567343, than in Daewonkong under waterlogging stress, as shown by qRT-PCR. The expression levels were normalized to that of *Act11* using a real-time polymerase chain reaction. DW: Daewonkong (light gray), PI 567343: waterlogging-resistant varieties (dark gray). ** *p* < 0.01; *** *p* < 0.001.

**Table 1 plants-13-02538-t001:** Soybean ortholog information of genes related to flooding stress identified in Arabidopsis and rice.

Plants	Initial_Alias (Gene_Id)	Symbol	Ortholog_Name	Description	DEG	Reference
Arabidopsis	*AT1G12010*	*ACO*	*Glyma.15G112700*	1-aminocyclopropane-1-carboxylate (ACC) oxidase	Down	[67]
	*AT1G12010*	*ACO*	*Glyma.09G008400*	1-aminocyclopropane-1-carboxylate (ACC) oxidase	Down	[67]
	*AT5G39890*	*AtPCO2*	*Glyma.19G020500*	cysteine oxidase (PCO)	Down	[69]
	*AT5G15120*	*AtPCO1*	*Glyma.19G020500*	cysteine oxidase (PCO)	Down	[69]
Rice	*Os07g47620*	*OsUsp1*	*Glyma.02G155600*	not annotated	Up	[70]
	*Os01g09700*	*OS-ACS5*, *OsACS5*	*Glyma.05G223000*	1-aminocyclopropane-1-carboxylate (ACC) synthase	Up	[71]
	*Os01g09700*	*OS-ACS5*, *OsACS5*	*Glyma.08G030100*	1-aminocyclopropane-1-carboxylate (ACC) synthase	Up	[71]
	*Os04g39020*	*BAD1*, *OsBADH1*	*Glyma.06G186300*	betaine-aldehyde dehydrogenase	Up	[72]
	*Os02g49720*	*ALDH2a*, *OsALDH2B5*	*Glyma.02G034000*	aldehyde dehydrogenase (ALDH2B)	Down	[68]

## Data Availability

The raw data of RNA-Seq analysis were deposited at ArrayExpress (E-MTAB-14196).

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
