# Peer review of "Characterization of the Regulatory Network under Waterlogging Stress in Soybean Roots via Transcriptome Analysis"

_plants, 2024, doi:10.3390/plants13182538_

Round 1

Reviewer 1 Report

Comments and Suggestions for Authors

In the enrichment analysis, how do you explain some biological pathways, such as amine biosynthetic process, alcohol metabolic process, can be enriched both for up-regulated and down-regulated genes. And I think GSEA analysis is better than GO enrichment.

Are hypocotyl roots relative to the waterlogging-tolerance of soybean, negative or positive ?

What is the meaning of the green box in Figure 4? 

Author Response

Response to Reviewer #1

Comment 1: In the enrichment analysis, how do you explain some biological pathways, such as amine biosynthetic process, alcohol metabolic process, can be enriched both for up-regulated and down-regulated genes. And I think GSEA analysis is better than GO enrichment.

Response to Comment 1:

We appreciate your constructive comment, as the suggestion, GSEA is a statistical method to analyze the function of gene sets. However, as you already know, the GSEA uses a full gene list to analyze functional categories. Our analysis applied GO enrichment to clarify the relationship between functional categories and expression change. In the case of the enriched GO terms in both DEGs, we illustrated them to reveal complex relativeness in the waterlogging response in soybean.

Comment 2: Are hypocotyl roots relative to the waterlogging-tolerance of soybean, negative or positive?

Response to Comment 2:

Hypocotyl roots act positively in waterlogging-tolerance as we described in the manuscript. (Lines 169-170). To improve clarity, we added and revised sentences in the manuscript (Lines 40-42 and 173-183). Thank you.

Comments 3: What is the meaning of the green box in Figure 4?

Response to Comment 3: The green box highlights the processes that mentioned in the manuscript (Lines 242-248). We added the information in Figure 3 legend (Line 262).

Reviewer 2 Report

Comments and Suggestions for Authors

In this manuscript, the authors presented data on the impact of waterlogging on the transcriptome in soybean. The results contribute to our understanding of soybean's resistance to waterlogging, but the dataset may not be sufficient for developing a new and compelling narrative.

Major points:

1.     Although it appeared that 7 figures were presented (including a workflow schematic that, in my opinion, is unnecessary), these figures could be condensed into 2-3 figures.

2.     The Results section was combined with some discussions, leading to the absence of the Discussion section. It is necessary to rewrite the Results and Discussion section.

3.     In figure 5B, the authors randomly selected some TF genes. In my opinion, it might be better to focus on the most significantly up- or down-regulated genes.

4.     In figure 5B, the expression of three ERFs (04G041200, 15G180000, 17G131900) was significantly decreased under waterlogging. It seems that these genes may play a negative role in resistance to waterlogging. However, in figure 7, the expression of these genes increased in the waterlogging resistant variety, suggesting they may play a positive role in waterlogging resistance. These results need to be thoroughly discussed.

Author Response

Response to Reviewer #2

In this manuscript, the authors presented data on the impact of waterlogging on the transcriptome in soybean. The results contribute to our understanding of soybean's resistance to waterlogging, but the dataset may not be sufficient for developing a new and compelling narrative.

Comment 1: Although it appeared that 7 figures were presented (including a workflow schematic that, in my opinion, is unnecessary), these figures could be condensed into 2-3 figures.

Response to Comment 1:

Thank you for your valuable comment. In the case of Figure 1, we maintained it as a Figure S1 to provide readers overall research workflow (Lines 85-96 and 383). Additionally, we rearranged the figures and merged Figure 6 and 7 to improve readability as the suggestion (Line 185, 232, 260, 291, and 333).

Comment 2: The Results section was combined with some discussions, leading to the absence of the Discussion section. It is necessary to rewrite the Results and Discussion section.

Response to Comment 2:

As suggested by the reviewer, we initially considered the separation of the Results and Discussion sections. However, we decided to combine them because incorporating interpretation and discussion with the transcriptome analysis result would be clearer to the readers. Also, there are instructions for Authors that the discussion part could be combined with the result section. We appreciate your thoughtful comment.

Comment 3: In figure 5B, the authors randomly selected some TF genes. In my opinion, it might be better to focus on the most significantly up- or down-regulated genes.

Response to Comment 3:

We agree with this comment. Focusing on the most significant TF genes, differential expression might be confirmed more clearly. However, we randomly selected TF genes to validate our DEG analysis because we thought the validation of randomly selected genes showed the non-biased result of our transcriptome analysis. Thank you for the suggestion.

Comment 4: In figure 5B, the expression of three ERFs (04G041200, 15G180000, 17G131900) was significantly decreased under waterlogging. It seems that these genes may play a negative role in resistance to waterlogging. However, in figure 7, the expression of these genes increased in the waterlogging resistant variety, suggesting they may play a positive role in waterlogging resistance. These results need to be thoroughly discussed.

Response to Comment 4:

Thank you for the constructive comment. As the reviewer pointed out, those three ERFs showed opposite expression between two soybean varieties (Daewonkong and PI 567343) under waterlogging stress. Based on the location of three ERFs in our co-functional network and refer previous report related to functional network under abiotic stress [1], it can be discussed that the opposite expression of the three ERFs indicates not just its negative or positive role but the existence of a co-functional network underlying waterlogging stress in soybean. We added discussion about this point in the manuscript (Lines 360-363).

Reference

  1. Urano, K.; Kurihara, Y.; Seki, M.; Shinozaki, K. ‘Omics’ Analyses of Regulatory Networks in Plant Abiotic Stress Responses. Current Opinion in Plant Biology 2010, 13, 132–138, doi:10.1016/j.pbi.2009.12.006.

Reviewer 3 Report

Comments and Suggestions for Authors

This study on the transcriptome analysis of soybean roots under waterlogging stress provides valuable insights into the molecular mechanisms of flooding tolerance in soybean. However, some improvements are required based on the comments to authors.

The abstract doesn't mention the number of upregulated vs. downregulated genes among the 1999 DEGs identified.

The duration of waterlogging stress (10 days) is quite long. It would be beneficial to include information on why this specific timepoint was chosen and whether earlier timepoints were also analyzed.

The abstract doesn't provide any specific examples of the identified genes or their potential functions, which could have strengthened the impact of the findings.

The abstract doesn't clearly state the criteria used for identifying DEGs (e.g., fold change and p-value cutoffs).

The statement "alcohol biosynthetic process" being the most significantly enriched GO term in downregulated genes is interesting but unexplained.

The transition between paragraphs could be smoother, particularly between the general introduction and the more specific discussion of flooding types and QTL studies.

In introduction add a clearer statement of the study's objectives.

Some sentences are quite long and could be split for better readability.

Paragraph 1 of the introduction could be cited with recent studies such as https://doi.org/10.3390/ijms22179175, DOI: 10.1016/j.plaphy.2021.01.042

Consider adding a brief mention of the physiological effects of waterlogging on plants to provide context for the molecular studies.

The description of QTL studies is quite detailed for an introduction. Consider summarizing this information more concisely.

The last paragraph (lines 75-84) reads more like a methods summary. Consider moving most of this to the methods section and replacing it with a concise statement of the study's objectives and approach.

There's a repetition in the section numbering: 2.3 and 2.4 are both titled "Functional analysis of DEGs using Gene Ontology (GO) and MapMan software".

The criteria for DEG selection (line 110-111) could be more clearly stated.

There's no mention of quality control steps for the RNA samples before library preparation.

Add information about RNA quality assessment (e.g., RNA integrity number) before library preparation.

Clarify whether the FPKM >= 4 criterion (line 111) is for both control and treatment conditions or just one.

In the last paragraph (lines 122-127), consider adding information about how the orthology between rice/Arabidopsis genes and soybean genes was determined.

Include some quantitative data on the extent of root growth inhibition and AR development, rather than just qualitative observations.

Add future perspective in conclusion section.  

Author Response

Response to Reviewer #3

This study on the transcriptome analysis of soybean roots under waterlogging stress provides valuable insights into the molecular mechanisms of flooding tolerance in soybean. However, some improvements are required based on the comments to authors.

Comment 1: The abstract doesn't mention the number of upregulated vs. downregulated genes among the 1999 DEGs identified.

Response to Comment 1:

To address this comment, we revised the DEG numbers in the abstract. (Line 17)

Comment 2: The duration of waterlogging stress (10 days) is quite long. It would be beneficial to include information on why this specific timepoint was chosen and whether earlier timepoints were also analyzed.

Response to Comment 2:

We appreciate this comment, as the reviewer suggested, 10 days of waterlogging stress seems quiet long period for soybean. However, based on previous reports on submergence and waterlogging stress treatment [1,2] and our morphological observation showing significant changes in root length and the number of adventitious roots after 10 days of waterlogging stress, we applied the period in this study. To provide more rational on the stress treatment, we added the sentences in the manuscript (Lines 101-102). 

Reference

  1. Koo, S.C.; Kim, H.T.; Kang, B.K.; Lee, Y.H.; Oh, K.W.; Kim, H.Y.; Baek, I.Y.; Yun, H.T.; Choi*, M.S. Screening of Flooding Tolerance in Soybean Germplasm Collection. Korean Society of Breeding Science 2014, 46, 129–135, doi:10.9787/KJBS.2014.46.2.129.
  2. Tamang, B.G.; Magliozzi, J.O.; Maroof, M. a. S.; Fukao, T. Physiological and Transcriptomic Characterization of Submergence and Reoxygenation Responses in Soybean Seedlings. Plant, Cell & Environment 2014, 37, 2350–2365, doi:10.1111/pce.12277.

Comment 3: The abstract doesn't provide any specific examples of the identified genes or their potential functions, which could have strengthened the impact of the findings.

Response to Comment 3:

Thank you for the suggestion, we have revised the abstract to increase readability and clarity for suggesting more specific information (Lines 16-25).

Comment 4: The abstract doesn't clearly state the criteria used for identifying DEGs (e.g., fold change and p-value cutoffs).

Response to Comment 4:

Thank you for the suggestion. Due to the word count limitation in the abstract, the criteria for DEG analysis are stated in the Method section (Lines 121-124).

Comment 5: The statement "alcohol biosynthetic process" being the most significantly enriched GO term in downregulated genes is interesting but unexplained.

Response to Comment 5:

Thank you for the suggestion. We have added explanations based on the reviewer’s comments (Lines 218-230).

Comment 6: The transition between paragraphs could be smoother, particularly between the general introduction and the more specific discussion of flooding types and QTL studies.

Response to Comment 6:

Thank you for the comment, we revised the paragraphs in the introduction section in increase readability (Lines 61-69).

Comment 7: In introduction add a clearer statement of the study's objectives.

Response to Comment 7:

Our objectives in this study are investigating transcriptome changes under waterlogging stress and identifying genetic resources that can be used for soybean breeding. To address the comment, we revised the sentences in the introduction to present our objectives in this study (Lines 79-83).

Comment 8: Some sentences are quite long and could be split for better readability.

Response to Comment 8:

During the revision, we split some sentences and rearranged them to increase readability (Lines 40-50, 79-83, 117-124, 175-183, 218-230, 355-364, 364-365, and 376-380). Thank you for the comment.

Comment 9: Paragraph 1 of the introduction could be cited with recent studies such as https://doi.org/10.3390/ijms22179175, DOI: 10.1016/j.plaphy.2021.01.042

Response to Comment 9:

Thank you for the suggestion, we strengthened our introduction by citing soybean omics research as reviewer’s suggestion (Line 61).

Comment 10: Consider adding a brief mention of the physiological effects of waterlogging on plants to provide context for the molecular studies.

Response to Comment 10:

To address the comment, we have added a brief summary of the physiological effects of flooding on plants in the Introduction section. (Lines 40-47).

Comment 11: The description of QTL studies is quite detailed for an introduction. Consider summarizing this information more concisely.

Response to Comment 11:

To address the comment, we condense the sentences of QTL studies to improve readability. Thank you for the comment (Lines 61-69).

Comment 12: The last paragraph (lines 75-84) reads more like a methods summary. Consider moving most of this to the methods section and replacing it with a concise statement of the study's objectives and approach.

Response to Comment 12:

We appreciate this comment. We rearranged the paragraph into Materials and Methods section (Lines 85-96) and added a new paragraph to reveal objective of this study in the end of the introduction section (Lines 79-83).

Comment 13: There's a repetition in the section numbering: 2.3 and 2.4 are both titled "Functional analysis of DEGs using Gene Ontology (GO) and MapMan software".

Response to Comment 13:

We apologize for confusing the reviewer. The title of 2.4 section was revised. (2.5 section in the revised manuscript, Line 134)

Comment 14: The criteria for DEG selection (line 110-111) could be more clearly stated.

Response to Comment 14:

To address the comment 4 and this comment, we updated the DEG selection process in the manuscript, especially average FPKM for filtering expressed gene (Line 123-124).

Comment 15: There's no mention of quality control steps for the RNA samples before library preparation.

Response to Comment 15:

To address this comment, we incorporated the quality control step information of the total RNA in the manuscript (Line 112-113).

Comment 16: Add information about RNA quality assessment (e.g., RNA integrity number) before library preparation.

Response to Comment 16:

Same as a response to comment 15, we incorporated the quality control information of the total RNA in the manuscript (Line 112-113).

Comment 17: Clarify whether the FPKM >= 4 criterion (line 111) is for both control and treatment conditions or just one.

Response to Comment 17:

Thank you for the comment, we applied the FPKM cut-off to identify actually expressed gene among DEGs. So, the criterion was applied for both control and treatment conditions. We updated the sentence at line 123

Comment 18: In the last paragraph (lines 122-127), consider adding information about how the orthology between rice/Arabidopsis genes and soybean genes was determined.

Response to Comment 18:

To address this comment, we revised the paragraph to integrate detailed ortholog analysis process (Lines 136-139)

Comment 19: Include some quantitative data on the extent of root growth inhibition and AR development, rather than just qualitative observations.

Response to Comment 19:

To address this comment, the quantitative data for root length and number of AR were incorporated into Figure 1 (Figure 1 D and E; Lines 187-188). We appreciate this suggestion.

Comment 20: Add future perspective in conclusion section.

Response to Comment 20:

Thank you for the suggestion. We intensively revised the conclusion paragraph to encompass the result of this study and future perspective (Lines 372-381).

Round 2

Reviewer 1 Report

Comments and Suggestions for Authors

No other questions!

Comments on the Quality of English Language

Quality of English Language is good.

Author Response

Response to Reviewer #1

No other questions!

Response to Reviewer 1: We appreciate your help in polishing our manuscript.

Reviewer 2 Report

Comments and Suggestions for Authors

The revised version of the manuscript looks a bit better than the original version. However, since the authors did not supply any new data, I believe that the data are not adequate to build a good story.

Author Response

Response to Reviewer #2

The revised version of the manuscript looks a bit better than the original version. However, since the authors did not supply any new data, I believe that the data are not adequate to build a good story.

Response to Reviewer 2: We appreciate your valuable comment. In this revision, we updated the qRT-PCR validation by adding two additional TFs, including the most significantly changed TF. In addition, we revised the discussion on the different expression patterns of three ERFs in two cultivars, referencing our previous study. We hope this revision will satisfy your suggestion.

Comment 3 from Reviewer2: In figure 5B, the authors randomly selected some TF genes. In my opinion, it might be better to focus on the most significantly up- or down-regulated genes.

Response to the Comment 3:

To address this question, we modified Figure 4 into revised Figure 4 and Figure S2 by adding qRT-PCR result of the two most highly changed TFs (Glyma.01g12970, bHLH; Glyma.18g252400, ERF). Furthermore, we discussed the most up or down-regulated genes in the manuscript (Lines 274-277 and 288-294). Thank you for the comment.

Comment 4 from Reviewer2: In figure 5B, the expression of three ERFs (04G041200, 15G180000, 17G131900) was significantly decreased under waterlogging. It seems that these genes may play a negative role in resistance to waterlogging. However, in figure 7, the expression of these genes increased in the waterlogging resistant variety, suggesting they may play a positive role in waterlogging resistance. These results need to be thoroughly discussed.

Response to the Comment 4:

In addition to the previous response, we discussed an example of a knockout mutant of a drought-responsive up-regulated gene that showed the opposite phenotype, drought tolerance [1](Lines 370-377). Taken together, functional validation needs to be performed to reveal the actual mode of action. We appreciate the constructive comment and hope this response satisfies the editor and reviewer.

Reference

  1. Yoo, Y.-H.; Nalini Chandran, A.K.; Park, J.-C.; Gho, Y.-S.; Lee, S.-W.; An, G.; Jung, K.-H. OsPhyB-Mediating Novel Regulatory Pathway for Drought Tolerance in Rice Root Identified by a Global RNA-Seq Transcriptome Analysis of Rice Genes in Response to Water Deficiencies. Front. Plant Sci. 2017, 8, doi:10.3389/fpls.2017.00580.

Round 3

Reviewer 2 Report

Comments and Suggestions for Authors

The revised manuscript looks a bit better than the last version. However, the main deficit still exists. It is difficult to build a novel and exciting story based on these preliminary data. Just analyzing the transcriptome data is not adequate.  I can not find any new mechanism in the manuscript.

It looks like the authors misunderstood my comments. Although they revised the formats and supplemented some data, the major problem of the manuscript was not solved. 

Author Response

Response to Reviewer 2:

We apologize that our revised manuscript didn’t satisfy your suggestion. Since our study was performed to identify genetic materials related to waterlogging stress in soybean plants, the description of the result was not focused on certain gene or mode of action. However, we tried to clarify the meaning of our transcriptome analysis and revised the manuscript. Thank you for your careful comment on our manuscript.